# KNOWLEDGE-ENHANCED MCTS FOR LLM-BASED MEDICAL DIAGNOSIS REASONING

## ABSTRACT

Medical diagnosis is a high-stakes, knowledge-intensive task that requires precise reasoning over complex patient information. While Large Language Models (LLMs) have shown promise across a range of medical applications, their ability to perform accurate and interpretable diagnostic reasoning remains limited. Existing LLM-based approaches often rely on shallow, single-step inferences and lack mechanisms to systematically evaluate multiple diagnostic hypotheses. To address these challenges, we propose Med-MCTS, a knowledge-enhanced diagnostic reasoning framework that integrates Monte Carlo Tree Search (MCTS) with external medical knowledge. Med-MCTS formulates diagnosis as a sequential decision-making process and introduces domain-specific state and action representations that align with clinical reasoning practices. During MCTS tree expansion, the model traverses structured medical knowledge graphs to enrich reasoning trajectories with relevant contextual information. To select high-quality paths, Med-MCTS employs a multi-dimensional scoring mechanism that evaluates self-consistency, factual accuracy, and diversity of reasoning. Experiments on multiple benchmark datasets demonstrate that Med-MCTS significantly improves diagnostic accuracy, enabling open-source LLMs to outperform domain-specific medical models and approach the performance of advanced proprietary systems such as GPT-4o.

## 1 INTRODUCTION

Large Language Models (LLMs) have revolutionized various fields, showcasing remarkable capabilities in natural language understanding and generation (Touvron et al., 2023; Yang et al., 2024). Recent advances in test-time scaling laws have revealed that extending the reasoning chain—by simulating the slow, deliberate thinking of human System 2 cognition—can significantly enhance LLM performance on complex problems (DeepSeek-AI et al., 2025; Li et al., 2025). Rather than relying on shallow, single-step inferences, recent models emphasize multi-step reasoning to explore and refine hypotheses in a systematic manner. This paradigm has led to notable progress in domains like mathematics, where structured reasoning aligns well with symbolic manipulation. Techniques such as Monte Carlo Tree Search (MCTS) have been successfully employed to decompose problems into discrete reasoning steps, build search trees, and optimize decision-making (Qi et al., 2024; Hao et al., 2023). These developments raise an important question: **Can similar tree-based reasoning strategies be applied effectively in high-stakes, knowledge-intensive domains like medicine**?

In clinical settings, robust reasoning is particularly critical during the diagnostic process. Medical diagnosis requires careful integration of heterogeneous data—structured inputs (e.g., lab results, demographics) and unstructured narratives (e.g., patient complaints, symptom history)—alongside external medical knowledge. Patients often do not exhibit the full clinical picture of a disease, presenting partial or atypical symptoms, further complicating the diagnostic task. Unlike mathematical problems, the reasoning in diagnosis is less about strict deductive chains and more about iterative hypothesis refinement informed by prior experience and domain expertise. Existing work that applies MCTS to medical diagnosis (Tran et al., 2024) typically adapts action decomposition strategies from mathematical reasoning. However, these methods fail to capture the nuanced, uncertain, and knowledge-rich nature of clinical decision-making. Physicians rely heavily on prior experience, background knowledge, and iterative testing of diagnostic hypotheses. This has strongly called for a reasoning framework that not only supports structured search but also accommodates the ambiguity and domain-specific subtleties inherent in medicine.

To address these challenges, we propose **Med-MCTS**, a knowledge-enhanced diagnostic reasoning framework that integrates Monte Carlo Tree Search with external medical knowledge. Med-MCTS models diagnosis as a sequential decision-making process with the following three characteristics tailored to the medical domain: (1) *Clinically-Informed Diagnostic Actions:* Unlike standard MCTS adaptations, which struggle with medical uncertainty, we design a hierarchical action space inspired by clinical reasoning patterns. This space consists of Key Symptom Extraction, Hypothesis Generation, Evidence Verification, and Deductive Analysis, explicitly modeling the iterative hypothesis-evaluation cycle of physicians. This design ensures interpretability and alignment with expert reasoning. (2) *Knowledge-Grounded Search Mechanism:* We incorporate medical knowledge graphs as verified reasoning substrates. By leveraging bidirectional symptom-disease relationships, the MCTS expansion process is guided by semantically meaningful and clinically accurate associations, improving diagnostic relevance and factual alignment. (3) *Multi-Dimensional Path Evaluation:* Med-MCTS introduces a comprehensive scoring mechanism that jointly considers logical consistency, factual accuracy, and reasoning path diversity. This evaluation approach enables the model to prioritize robust and plausible diagnostic trajectories, better reflecting the uncertainty and hierarchy of real-world clinical reasoning.

Our experiments on multiple benchmark datasets demonstrate that Med-MCTS significantly improves diagnostic accuracy. In particular, Med-MCTS enables open-source LLMs to outperform domain-specific medical models and approach the performance of advanced proprietary systems, such as GPT-4o, in terms of diagnostic quality. These findings demonstrate the potential of structured, knowledge-enhanced reasoning to advance trustworthy LLMs in healthcare.

## 2  RELATED WORK

**Language Models in Clinical Diagnosis.**    Timely and accurate diagnosis is foundational to effective clinical care and is the critical first step in ensuring appropriate patient outcomes (Singh et al., 2019). The diagnostic process typically involves integrating a patient's medical history, physical signs, and other clinical information to identify potential diseases. With recent advances in large language models, there is growing interest in leveraging these models to support diagnostic reasoning. Existing medical LLMs typically follow two main approaches: **(1) Prompt Engineering-Based Approach** (Saab et al., 2024; Chen et al., 2024e; Nori et al., 2023; Li et al., 2024; Kim et al., 2024; Tang et al., 2023): This approach carefully designs prompts that incorporate patient symptoms, medical history, and contextual cues to guide the model's responses. It allows for flexible inference and rapid hypothesis generation even with sparse input data. Although efficient and training-free, it struggles to adapt to the complexity of clinical diagnostic scenarios. **(2) Fine-Tuning with Medical Data** (Chen et al., 2024c; Wang et al., 2025a;c; Labrak et al., 2024; Tian et al., 2023; Christophe et al., 2024): This approach involves domain-specific pretraining or fine-tuning on large-scale medical corpora, enabling deeper understanding of medical concepts and clinical patterns. However, this method is resource-intensive and struggles to keep up with the rapidly evolving landscape of medical knowledge (Labrak et al., 2024; Zhou et al., 2023).

**Test-Time Scaling Law.**    The Test-Time Scaling Law refers to enhancing LLM's performance during the inference stage by increasing computational resources or inference time (Snell et al., 2024). The simplest implementation involves increasing the number of generations, such as sampling multiple candidate responses using Chain of Thought (Wei et al., 2023) and then selecting the best answer (Best-of-N) using a predefined discrimination strategy (Lightman et al., 2023; Uesato et al., 2022; Wang et al., 2023; Madaan et al., 2023; Shinn et al., 2023). However, simple multi-sampling approaches are constrained by limited search diversity and heavily rely on the reliability of scoring functions. To address this, recent methods explore richer reasoning trajectories via tree-based search over decomposed thought steps (Qi et al., 2024; Hao et al., 2023; Yao et al., 2023; Markowitz et al., 2024; Xie et al., 2023; Chen et al., 2024a). Traditional tree-based search methods, such as beam search (Xie et al., 2023) and depth-/breadth-first search (Yao et al., 2023), can explore non-linear reasoning structures but are limited in path optimization. To address this, Monte Carlo Tree Search (MCTS) (Qi et al., 2024; Hao et al., 2023; Chen et al., 2024a) evaluates the potential value of different paths through random sampling, enabling more efficient exploration of the search space and improving the accuracy and efficiency of reasoning.

# 3 TASK FORMULATION AND PRELIMINARIES

## 3.1 FORMULATION OF MEDICAL DIAGNOSIS

Medical diagnosis is the process of identifying and determining the nature of a disorder or illness through comprehensive evaluations of a patient's symptoms, medical history, and physical examination findings. It can be formalized as a function $f$ that maps a patient's feature set to a specific disease $d$: $d = f(\mathcal{P}, \mathcal{C})$. $\mathcal{P} = \{p_0, p_1, \ldots p_{n_p}\}$ represents the patient's general features (e.g., age, gender, and past medical history), where $n_p$ is the number of such features. $\mathcal{C} = \{c_0, c_1, \ldots, c_{n_c}\}$ denotes the clinical features (e.g., symptoms and test results), where $n_c$ is the size of the set. These features serves as critical evidence for the doctor's diagnosis. The output $d \in \mathcal{D}$ corresponds to the diagnostic conclusion where $\mathcal{D}$ denotes the set of all diseases.

We aim to reconstruct the clinical diagnostic reasoning using language models, with the dual objectives of **(1)** achieving accurate medical diagnosis and **(2)** generating interpretable reasoning chains to enhance the transparency and credibility of diagnostic outputs. To achieve this objective, we systematically analyze the cognitive reasoning mechanisms employed by physicians during clinical diagnosis (Garibaldi & Olson, 2018)(Wolf, 1985). A standard diagnostic process typically comprises the following key steps:

**A. Medical History Taking.** In real-world scenarios, the input to the medical diagnosis problem is typically a patient's verbal description $x$. A pre-processing function $f_{\text{sum}} : x \to (\mathcal{P}, \mathcal{C})$ summarizes the general features and clinical characteristics of the patient.

**B. Hypothesis Generation.** A hypothesis generator $f_{\text{hypo}} : (\mathcal{P}, \mathcal{C}) \to \mathcal{D}_{\text{hypo}}$ produces an initial set of possible disease hypotheses $\mathcal{D}_{\text{hypo}} \subseteq \mathcal{D}$ based on the observed patient features.

**C. Hypothesis Characterization.** For each hypothesis disease $d_h \in \mathcal{D}_{\text{hypo}}$, an inference process $f_{\text{ret}} : d_h \to (\mathcal{P}_h, \mathcal{C}_h)$ retrieves the features $\mathcal{P}_h$ and $\mathcal{C}_h$ associated with $d_h$. This provides expected manifestations of each hypothesis and determines if further testing is needed to support diagnosis.

**D. Priority Assessment and Decision-Making.** An evaluation function $f_{\text{eval}} : (\mathcal{P}_h, \mathcal{C}_h) \times (\mathcal{P}, \mathcal{C}) \to \rho_h$ computes the degree of similarity between the features of patient and the expected features under hypothesis $d_h$. The result is a rating score $\rho_h \in [0, 1]$ for each hypothesis. The final diagnostic conclusion is the hypothesis with the highest rank: $d = \arg\max_{d_h \in \mathcal{D}_{\text{hypo}}} \rho_h$.

## 3.2 MONTE CARLO TREE SEARCH

Medical diagnosis can be decomposed into a multi-step reasoning problem, modeled as a search tree constructed by MCTS. Specifically, given an initial state $s_0$, which in this context represents the patient's description $x$, the construction of search tree $\mathcal{T}$ can be formulated as an iterative expansion process. As shown in Figure 2, each node of $\mathcal{T}$ represents a state $s \in \mathcal{S}$ and each edge represents an action $a \in \mathcal{A}$. During each decision step $t$, the model must select an optimal action $a_t = \pi_{\text{LLM}}(s_t)$ and transition from $s_t$ the subsequent state $s_{t+1}$. A path from the root node to the leaf node is a candidate reasoning trajectory $t = x \oplus s_1 \oplus s_2 \oplus \cdots \oplus s_{leaf}$. Ultimately, we sample multiple trajectories as a set $\mathbb{T} = \{t_0, t_1, \ldots, t_{n_t}\}$ from the tree $\mathcal{T}$ and derive the final answer through a specific strategy, where $n_t$ represents the total number of sampled trajectories.

Monte Carlo Tree Search is a search algorithm that combines the precision of tree search with the generality of random sampling (Browne et al., 2012). The algorithm constructs and expands the search tree through four key steps: **Selection**—using a heuristic strategy to traverse the reasoning tree and select a leaf node; **Expansion**—adding child nodes to increase the search space; **Simulation**—starting from the newly expanded node, performing a random simulation (a rollout) until reaching a terminal state; **Backpropagation**—propagating the simulation results backward from the leaf to the root node, updating the statistical information of all nodes along the path to guide future searches.

While MCTS works well in math by treating each step as a clear state or action (Qi et al., 2024; Hao et al., 2023), applying it to medicine is more difficult. Medical diagnosis involves complex and unclear situations that are hard to formalize. The reasoning process depends heavily on physicians' knowledge and experience, making it hard to break down into simple steps. As a result, methods used to decompose math problems (Tran et al., 2024) often do not fit medical tasks, leading to reasoning that is hard to interpret in clinical practice.

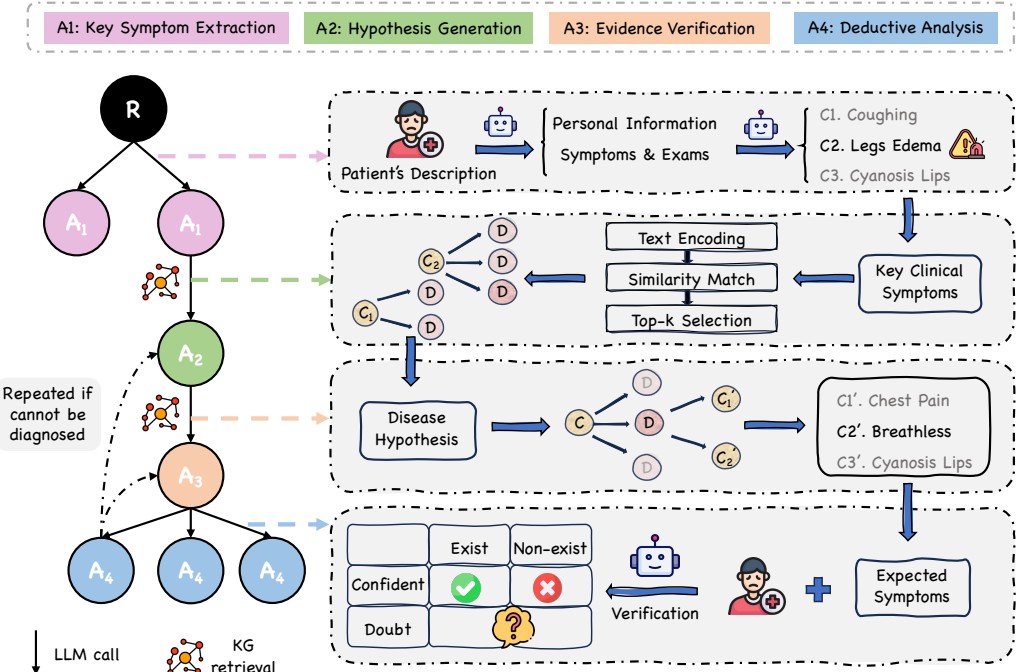

Figure 1: The Med-MCTS framework implements four actions: (A1) Key Symptom Extraction identifies critical symptoms from patient complaints; (A2) Hypothesis Generation retrieves potential diseases from medical knowledge graphs using extracted symptoms; (A3) Evidence Verification performs reverse knowledge graph queries to obtain disease-specific symptoms; and (A4) Deductive Analysis evaluates evidence to validate diagnostic hypotheses, with automatic backtracking to A2/A3 for implausible cases.

## 4 METHOD

This section presents the Med-MCTS framework, with actions defined in Subsection 4.1, knowledge-enhanced search in Subsection 4.2 and path evaluation in Subsection 4.3. The overall Med-MCTS algorithm is presented in Figure 1 and Algorithm 1.

### 4.1 CLINICALLY-INFORMED DIAGNOSTIC ACTIONS

As discussed before, the action and state space employed in mathematics-MCTS cannot be directly transferred to medical diagnostic tasks. To address this, we introduce a hierarchical action space that more closely mimics real-world medical scenarios. The proposed action space is defined as follows:

⋄**A1: Key Symptom Extraction.** To address the inherent variability of real-world clinical presentations—often accompanied by substantial irrelevant or misleading information—A1 is designed to extract key symptoms from complex and unstructured patient descriptions. By filtering out noise at the first level of the reasoning tree, this action distills core clinical signals, enabling MCTS to perform a systematic and targeted exploration of diagnostic hypotheses based on critical evidence.

⋄**A2: Hypothesis Generation.** Building upon the symptoms extracted in A1 and incorporating the patient's medical history and general information, this action generates a potential disease hypothesis for the second level nodes of the reasoning tree, augmented by external knowledge.

⋄**A3: Evidence Verification.** This action retrieves disease-specific symptom profiles from external knowledge sources based on the initial diagnostic hypothesis and identifies the most clinically significant indicators requiring validation. Operating at the third level of the reasoning tree, this action emulates physicians' reasoning processes in differential diagnosis.

⋄ *A4: Deductive Analysis.* This action analyzes the results of evidence verification to assess whether the patient's presentation supports the current hypothesis, whether further testing is needed, or whether the hypothesis should be refuted. This includes:

- Exist and Confident: The clinical indicator requiring validation exists and support the current hypothesis, leading to a final diagnosis.

- Exist but Doubt: The indicator requiring validation exists but uncertainty remains. Further diagnostic tests are required to confirm the hypothesis, i.e., return to A3.

- Non-exist and Confident: The indicator requiring validation does not exists, thus refuting the current hypothesis. The model should consider other possible diseases, i.e., return to A2.

- Non-exist but Doubt: Although the indicator requiring validation does not exist, other clinical findings or auxiliary information may still support the current hypothesis. The model should perform deeper pathological analysis to explain this divergence and trigger additional diagnostic verification, i.e., return to A3.

By introducing this hierarchical and more realistic medical scenario-based action space, Med-MCTS can simulate a semi-interactive diagnostic process. Starting from the root node $s_0$, we perform selection, expansion, simulation, and backpropagation to construct a complete medical reasoning tree. During the selection phase, we employ the Upper Confidence Bound apply to Tree (UCT) (Kocsis & Szepesvári, 2006) to balance exploration and exploitation. The formula for UCT is:

$$UCT(s, a) = \bar{Q}(s, a) + c\sqrt{\frac{\ln N_{parent}(s)}{N(s, a)}}, \tag{1}$$

where $\bar{Q}(s, a) = \frac{Q(s,a)}{N(s,a)}$ denotes the average reward obtained of action $a$ in state $s$, with $Q(s, a)$ as the estimated reward value. $N(s, a)$ is the number of times node $s$ has been visited, and $N_{parent}(s)$ indicates the visit count of the parent node of $s$.

## 4.2 KNOWLEDGE-GROUNDED SEARCH MECHANISM

Medical diagnosis depends heavily on domain-specific knowledge. However, MCTS-based diagnostic methods (Qi et al., 2024; Tran et al., 2024) exhibit significant limitations in integrating external knowledge: these approaches either overly rely on the model's internal parametric knowledge or merely employ retrieval-augmented generation (RAG) mechanisms based solely on document similarity. Such simplistic text-matching methods demonstrate notable shortcomings in complex medical diagnostic scenarios. First, medical texts are highly specialized and complex, making surface-level similarity-based retrieval inadequate for accurately identifying subtle distinctions between medical terminologies. Second, the diagnostic process requires establishing logical relationships among symptoms, test results, and diseases, whereas traditional RAG methods depend on text fragment matching, failing to capture such deep-level logical connections. In contrast, knowledge graphs offer clear advantages by representing medical knowledge in a structured form, including relationships between diseases, symptoms, test results, and treatments. Additionally, models can leverage paths within the graph for causal reasoning, enabling step-by-step derivation of diagnostic results. This provides more transparent diagnostic evidence, enhancing both the accuracy and interpretability of diagnostic systems.

Based on the above analysis, we propose a knowledge-enhanced approach that integrates medical knowledge graphs into MCTS. Specifically, given a medical knowledge graph $\mathcal{G} = \{V, E\}$, we first encode all nodes $v \in V$ using a text embedding model $f_\theta$, constructing an entity vector space $\{\mathbf{v}_i = f_\theta(v_i) | v_i \in V\}$. During inference, for a given query text $q$ and target relation type $R$, we employ a large language model to parse $q$ into a set of medical entities $\mathcal{E}_q = \{e_0, e_1, \dots, e_{n_q}\}$ and embed these entities into corresponding query vectors $\{\mathbf{e}_i = f_\theta(e_i)\}_{i=0}^{n_q}$ using the same model, where $n_q$ denotes the total number of extracted medical entities. Next, we compute the cosine similarity between each query entity and all nodes in $\mathcal{G}$, and selecte the top-$k$ matching entities with cosine similarity scores,

$$\mathcal{V}_i = \underset{v_j \in V, |S|=k}{\arg \text{top-}k} \ \cosine(e_i, v_j). \tag{2}$$

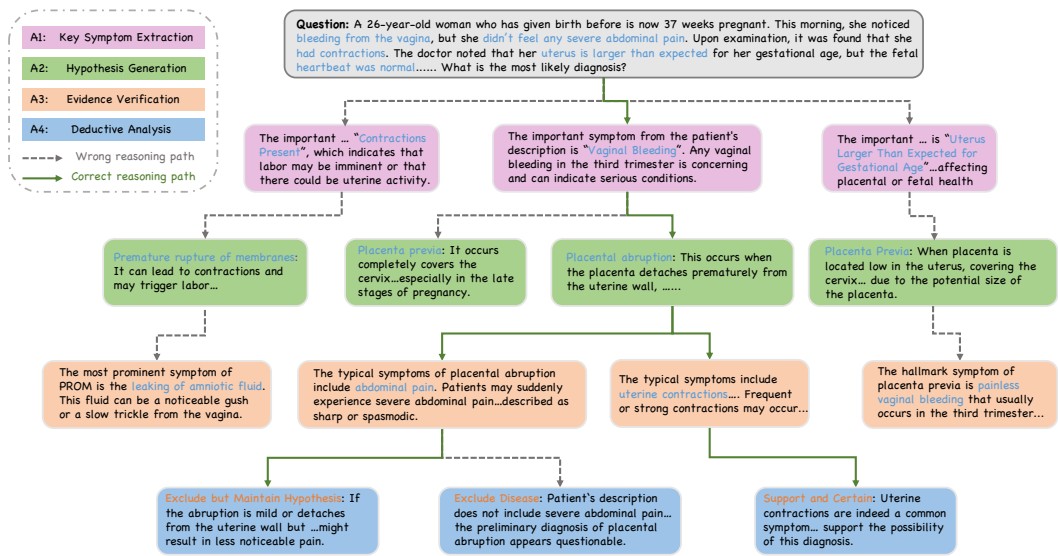

Figure 2: A case of Med-MCTS workflow for answering the question sampled from MedQA.

To ensure the reliability of retrieved knowledge, we establish a similarity threshold $\tau \in [0, 1]$. If $\max_j \cos(e_i, v_j) < \tau$, the system automatically disables knowledge augmentation for this query, using only its internal parametric knowledge. This threshold mechanism serves as a quality control that prevents the integration of potentially irrelevant or low-confidence knowledge from the external knowledge graph $\mathcal{G}$. The subgraph $\mathcal{G}_{sub} \subseteq \mathcal{G}$ under the relation type $R$ is retrieved as external knowledge augmentation,

$$\mathcal{G}_{sub} = \bigcup_{i=1}^{n} \{(v_h, rel, v_t) \in E | v_h \in \mathcal{V}_i, rel \in R\}. \tag{3}$$

Our framework employs two complementary retrieval methods from the medical knowledge graph to support differential diagnosis:

$\diamond$ **R1: Symptom-to-Disease Retrieval.** Given a patient's clinical symptom set $\mathcal{C} = \{c_1, \ldots, c_{n_c}\}$, we query the knowledge graph $\mathcal{G}$ to retrieve potential disease candidates $\mathcal{D}_{cands} = \{d \mid \exists c \in \mathcal{C}, (c, \text{causes}, d) \in \mathcal{G}\}$ ranked by clinical association strength. This forward-chaining reasoning is primarily applied during the initial tree expansion phase to generate plausible diagnostic hypotheses for further evaluation.

$\diamond$ **R2: Disease-to-Symptom Retrieval.** For each disease hypothesis $d_i \in \mathcal{D}_{hypo}$, we perform inverse retrieval to obtain its characteristic clinical manifestations $\mathcal{C}_{d_i} = \{c \mid \exists d \in \mathcal{D}_{hypo}, (d, \text{manifests}, c) \in \mathcal{G}\}$, including typical symptoms, pathological mechanisms, and potential complications. This backward-chaining verification serves to assess the congruence between observed symptoms and disease profiles, and guide the selection of subsequent diagnostic tests through the relation $(d, \text{manifests}, c) \in \mathcal{G}$.

The bidirectional retrieval mechanism ensures comprehensive differential diagnosis while maintaining clinical interpretability.

### 4.3 MULTI-DIMENSIONAL PATH EVALUATION

At the termination of the exploration phase in MCTS, we designed a comprehensive evaluation method to select the optimal diagnostic result from the multiple candidate trajectories. Inspired by evaluation mechanisms used in (Qi et al., 2024; Lifshitz et al., 2025), this method integrates multi-dimensional evaluators to assess answer consistency, reasoning path diversity, and factual accuracy. Specifically, we first sample all complete reasoning trajectories $\mathbb{T} = \{t_0, t_1, \ldots, t_{n_p}\}$ from the tree, then classify these paths into corresponding trajectory sets $\{A_0, A_1, \ldots, A_{n_\alpha}\}$ based on their final

Table 1: Answer accuary (%) of our Med-MCTS and other baseline methods on four medical diagnostic benchmarks. SC is self-consistency. @maj uses majority voting for answer verfication.

| Model | Method | Datasets | | | |
|---|---|---|---|---|---|
| | | MedBullets-4op | MedBullets-5op | MedQA | JMED |
| Qwen2.5-7B-instruct | CoT | 42.97 | 35.71 | 64.31 | 53.34 |
| | RAG | 48.70 | 38.64 | 69.13 | 54.34 |
| | SC@maj8 | 46.43 | 37.34 | 64.95 | 55.00 |
| | RAP | 48.38 | 40.26 | 65.92 | 56.67 |
| | rStar | 49.03 | 41.56 | 69.45 | 57.33 |
| | **Med-MCTS** | **52.27** | **43.18** | **72.67** | **58.67** |
| Qwen2.5-72B-instruct | CoT | 63.31 | 54.55 | 76.53 | 64.00 |
| | RAG | 67.86 | 55.52 | 80.71 | 64.33 |
| | SC@maj8 | 65.91 | 55.84 | 78.78 | 64.67 |
| | RAP | 66.56 | 57.14 | 81.99 | 66.33 |
| | rStar | 68.18 | 58.77 | 82.96 | 67.00 |
| | **Med-MCTS** | **71.10** | **62.66** | **84.57** | **68.00** |

diagnostic results $\{\alpha_0, \alpha_1, \ldots, \alpha_{n_\alpha}\}$, where $n_\alpha$ represents the total number of distinct diagnostic results. For each answer category $\alpha_k$, its evaluation score consists of the following components:

$$\text{Score}(\alpha_k) = \underbrace{\lambda_1 \frac{|A_k|}{m}}_{\text{Consistency}} - \underbrace{\lambda_2 \frac{\sum_{t_i \in A_k} \sum_{t_j \in A_k, t_j \neq t_i} \text{Sim}(t_i, t_j)}{|A_k|(|A_k| - 1)}}_{\text{Diversity}} + \underbrace{\lambda_3 \frac{\sum_{t_i \in A_k} \text{MAE}(t_i)}{|A_k|}}_{\text{Agent Evaluation}}. \quad (4)$$

**(1) Consistency** measures the frequency of the answer appearing across all reasoning paths. High consistency indicates that the answer is supported by the majority of reasoning paths, reflecting its stability and reliability. **(2) Diversity** evaluates the variability among all reasoning paths leading to the answer. High diversity indicates that the answer can be supported through multiple distinct reasoning paths, reflecting the comprehensiveness and flexibility of the diagnostic process. **(3) Agent Evaluation** assesses the factual accuracy of the answer by simulating the evaluation in a multi-agent system. $\lambda_1$, $\lambda_2$ and $\lambda_3$ are tunable hyperparameters that control the weight for each component. Together, these components provide a robust and multi-dimensional evaluation for selecting the optimal diagnostic result.

## 5 EXPERIMENTS

### 5.1 SETUP

We evaluate our proposed method, Med-MCTS, on four medical diagnosis benchmarks—MedQA, Medbullets-4options, Medbullets-5options, and JMED—using two open-source LLMs: Qwen2.5-7B-Instruct and Qwen2.5-72B-Instruct (Yang et al., 2024).

**Baselines.** Med-MCTS is a slow-thinking framework for various LLMs. We consider the following baselines: **Chain-of-Thought (CoT)** (Wei et al., 2023) Prompting guides the model to generate a series of intermediate reasoning steps by providing a few demonstrations. **Retrieval-Augmented Generation (RAG)** (Lewis et al., 2021) enhances model's knowledgeability and accuracy by retrieving relevant information from an external knowledge base and integrating it into the generation process. **Self-Consistency (SC)** (Wang et al., 2023) leverages the intuition that a complex reasoning problem typically admits multiple different ways of thinking leading to its unique correct answer. **RAP** (Hao et al., 2023) adopts a self-exploration solution to iteratively improve LLM's reasoning performance through self-rewarded feedback. **rStar** (Qi et al., 2024) advocates a richer set of reasoning actions and augments the MCTS process with a mutual consistency discrimination process. In addition, we conducted comparative evaluations with frontier general-purpose LLMs, including GPT-4o (OpenAI, 2024a), o1-preview (OpenAI, 2024b), DeepSeek-v3 (DeepSeek-AI, 2025) and DeepSeek-R1 (Guo et al., 2025), as well as domain-specific models specifically trained on the medical data, including HuatuoGPT-o1 (Chen et al., 2024d) and Citrus1.0-Qwen (Wang et al., 2025b).

## 5.2 Main Results

As shown in Table 1, this study first systematically compares Med-MCTS with other methods based on reasoning cost expansion, including CoT, SC, RAP, and rStar. As Med-MCTS is a model-agnostic, plug-and-play reasoning enhancement framework, we evaluate its performance across backbone models of varying scales to validate its generalizability. The experimental results demonstrate that, when using the same backbone model, Med-MCTS achieves state-of-the-art performance across all four benchmark datasets. This advantage primarily stems from its innovative integration of reasoning actions with knowledge graph retrieval augmentation. Specifically, on the Qwen2.5-7B model, Med-MCTS achieves performance improvements of 3.24%, 1.62%, 3.22%, and 1.34% on MedBullets-4op, MedBullets-5op, MedQA, and JMED datasets compared to rStar. Similarly, for the Qwen2.5-72B model, Med-MCTS exhibits substantial performance gains, surpassing rStar by 2.92% on MedBullets-4op, 3.89% on MedBullets-5op, 1.61% on MedQA, and 1.00% on JMED. These consistent improvements highlight the effectiveness of the Med-MCTS approach across different model scales. Med-MCTS demonstrates stable performance improvements independent of model scale. Moreover, on certain datasets, its reasoning capability improves more substantially as the model size increases (from 7B to 72B), indicating better adaptability to larger models. Notably, even when compared against RAG baselines that also employ external knowledge augmentation, Med-MCTS exhibits significant performance improvements, which strongly validates the unique value of knowledge graphs in medical diagnostic tasks.

Table 2 compares the performance of current SOTA general-purpose commercial models and specialized medical models. The results show that, despite Med-MCTS's inherent limitations as a training-free prompt-based method, its version equipped with a 72B general-purpose open-source model still surpasses all specialized medical models and approaches the performance level of GPT-4o. Due to the constraints in both model size and training scale of our base model, there remains a performance gap compared to the SOTA results achieved by GPT-o1-preview. Combined with the model-scale correlation observed in Table 1, we can reasonably hypothesize that Med-MCTS could achieve even better performance if integrated with larger-scale, domain-specialized backbone models possessing stronger reasoning capabilities.

Table 2: Answer accuracy (%) of frontier LLMs methods on four medical diagnostic benchmarks.

| Model | Datasets | | | | |
|---|---|---|---|---|---|
| | MedBullets (4op) | MedBullets (5op) | MedQA | JMED | Overall |
| *Frontier-level General LLMs* | | | | | |
| DeepSeek-v3 | 61.69 | 56.82 | 78.14 | 64.60 | 65.31 |
| DeepSeek-R1 | 81.82 | 68.51 | 90.68 | 68.70 | 77.43 |
| GPT-4o | 75.00 | 71.10 | 82.32 | 62.70 | 72.78 |
| GPT-o1-preview | **87.62** | **83.77** | **94.21** | **71.60** | **84.30** |
| *Medical-specific LLMs* | | | | | |
| HuatuoGPT-o1-7B | 51.62 | 40.48 | 69.77 | 54.50 | 54.09 |
| HuatuoGPT-o1-72B | 74.03 | 61.34 | 83.60 | 65.80 | 71.19 |
| Citrus1.0-Qwen-72B | 66.23 | 55.52 | 88.75 | 68.90 | 69.85 |
| **Med-MCTS** (Ours) | 71.10 | 62.66 | 84.57 | 68.00 | 71.58 |

Table 3: Ablation study on Med-MCTS components

| Settings | Accuray(%) |
|---|---|
| *Analysis of knowledge-enhanced Modules* | |
| Disable R1 | 71.7 |
| Disable R2 | 70.0 |
| Disable R1&R2 | 66.7 |
| *Analysis of deductive analysis Modules* | |
| Disable A4 | 68.3 |
| *Analysis of discriminator effectiveness* | |
| Majority vote | 64.2 |
| Mult-Agent evalution | 67.5 |
| Enable All | 72.5 |

## 5.3 Ablation Study and Analysis

**Effectiveness of the Med-MCTS Generator.** To systematically evaluate the efficacy of each module in the Med-MCTS generator, we conducted ablation experiments on 120 samples from the MedQA dataset using the Qwen2.5-7B model. Table 3 presents the accuracy results under different configurations. As shown in the upper section of Table 3, the experimental results demonstrate that removing the knowledge graph retrieval augmentation module (R1/R2) leads to a significant accuracy drop of 5.8 % (from 72.5% to 66.7%). This strongly validates the critical role of external medical knowledge enhancement in clinical diagnostic reasoning. Furthermore, introducing the deductive analysis action (A4) improves accuracy by 4.2%. This enhancement confirms that relying solely on symptom presence/absence for diagnosis is insufficient in complex clinical scenarios. The A4 module effectively improves the model's judgment capability for challenging cases by simulating physicians'

reflective reasoning processes. The complete Med-MCTS framework—integrating knowledge graph retrieval augmentation, deductive analysis, and the multi-dimensional discriminator—achieves the best performance 72.5%. This result robustly demonstrates the synergistic effects of these modules.

**Effectiveness of the Med-MCTS Discriminator.** To verify the efficacy of the proposed discriminator, we compared it with two baseline methods: majority voting and multi-agent verification (Lifshitz et al., 2025). As illustrated in the lower section of Table 3, when validating answers based on the same reasoning trajectories from the generator, our multi-dimensional discriminator exhibits significant advantages in reasoning accuracy. This performance improvement primarily stems from our novel hybrid discrimination strategy, which innovatively combines objective assessment (based on statistical analysis) and subjective verification (based on model self-reflection). Compared to traditional single-mechanism approaches, this dual-track verification framework more comprehensively captures critical information in the reasoning process, effectively reducing misjudgment risks.

**Expert Evaluation and Broader Implications** The multiple-choice format datasets provide unambiguous ground-truth answers, thereby avoiding the noise inherent in automated metrics—such as variations due to entity linking (e.g., synonyms for drugs or diseases) or differences in phrasing—and enables a more precise evaluation of diagnostic reasoning capabilities. While we recognize that automatic evaluation alone may not fully reflect a model's clinical utility, we conducted an expert evaluation to complement the automatic results. Three medical graduate students independently scored 50 randomly sampled diagnostic paths generated by Med-MCTS on a 5-point scale across two criteria: diagnostic correctness and interpretability. The mean scores were 4.12 and 4.23, respectively. These results provide additional evidence of the potential of Med-MCTS in handling realistic clinical scenarios with unstructured inputs. Beyond outputting a final answer, the reasoning tree exposes the entire decision path—key symptom → candidate disease → evidence supplementation → diagnosis refinement. This transparency provides multifaceted value that a single accuracy metric cannot capture. For clinicians, it offers a verifiable "show-your-work" pathway, building trust and allowing for rapid identification of potential flaws. For system improvement, the tree's branches naturally yield high-quality preference data for future alignment. For long-term maintainability, the knowledge-grounded approach allows the system to evolve with medical knowledge. Consequently, the expert-evaluated interpretability confirms that the primary benefit of Med-MCTS lies in delivering actionable and auditable reasoning, which is paramount for high-stakes clinical decision support.

## 6   CONCLUSION

In this work, we present Med-MCTS, a knowledge-enhanced reasoning framework specifically designed for medical diagnostic tasks. As a plug-and-play reasoning enhancement, Med-MCTS can be seamlessly integrated with various large language models without requiring additional fine-tuning or training. By designing clinically-informed diagnostic actions, incorporating external medical knowledge graphs, and developing a multi-dimensional evaluation mechanism, our framework significantly improves both the accuracy and interpretability of medical diagnostic reasoning. Experimental results demonstrate that Med-MCTS achieves outstanding performance across multiple medical reasoning tasks, outperforming existing baseline methods and reaching performance levels comparable to GPT-4o, providing a novel solution for AI-assisted medical diagnosis.

## LIMITATIONS

A primary limitation of Med-MCTS lies in its computational overhead relative to standard reasoning methods. The tree-search process inherently requires multiple LLM calls, leading to increased inference latency and resource consumption compared to a single-pass generation. This cost-effectiveness trade-off may currently limit its applicability in scenarios demanding real-time, low-cost interactions. However, we argue that in high-stakes domains like medical diagnosis, this cost is justifiable. The critical imperative is diagnostic accuracy and the provision of a verifiable reasoning trace, both of which are enhanced by the systematic exploration of Med-MCTS. The potential human and financial costs of a misdiagnosis far outweigh the incremental computational expense incurred by our method. Therefore, while future work will focus on optimizing efficiency, we believe the Med-MCTS approach represents a critical trade-off in favor of safety and reliability for healthcare applications.

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

## A  ALGORITHM OF MED-MCTS

---

**Algorithm 1** Workflow of Med-MCTS

---

1: **Input:** Patient's verbal description $x$, $LLM$, Knowledge Graph $\mathcal{G}$, rollout number $k$.
2: $(\mathcal{P}, \mathcal{C}) \leftarrow$ MedExtractor $LLM(x)$
3: Initialize Tree $\mathcal{T}$ using $(\mathcal{P}, \mathcal{C})$
4: **for** $i = 1$ **to** $k$ **do**
5:     node $s \leftarrow$ select($\mathcal{T}$)
6:     $S_{\text{child}} \leftarrow$ expand($\mathcal{T}$, $s$)
7:     Randomly sample a node $s_{\text{sample}}$ from $S_{\text{child}}$
8:     $s_{\text{sol}} \leftarrow$ SIMULATE($s_{\text{sample}}, \mathcal{P}, \mathcal{C}$)
9:     Backpropagate($\mathcal{T}$, $s_{\text{sol}}$)
10: **end for**
11: Sample all trajectories $\{t_0, t_1, \ldots, t_{n_t}\}$ and group them by final answer $\{\alpha_0, \alpha_1, \ldots, \alpha_{n_\alpha}\}$
12: **return** $\alpha_{\text{best}} = \arg\max(Score(\alpha))$

13: **function** SIMULATE($s, \mathcal{P}, \mathcal{C}$)         ▷ Taking simulation starting from root node as example
14:     **(A1)** $c \leftarrow$ Extractor of key symptoms $LLM(\mathcal{P}, \mathcal{C})$
15:     **(R1)** Retrieve candidate diseases $\mathcal{D}_{\text{cands}} \leftarrow (c, \mathcal{G})$
16:     **(A2)** $d_h \leftarrow$ Generator of hypotheses $LLM(c, \mathcal{D}_{\text{cands}}, \mathcal{P})$
17:     **(R2)** Retrieve clinical manifests $\mathcal{C}_{\text{hypo}} \leftarrow (d_h, \mathcal{G})$
18:     **(A3)** $\hat{c} \leftarrow$ Selector of related symptoms $LLM(c, \mathcal{C}_{\text{hypo}}, \mathcal{P})$
19:     **(A4)** $s_{\text{sol}} \leftarrow$ Generator of diagnosis $LLM(\hat{c}, d_h, \mathcal{P}, \mathcal{C})$
20:     **if** $s_{\text{sol}}$ confirms $d_h$ **then**                         ▷ Exist and confident
21:         **return** $s_{\text{sol}}$
22:     **else if** $s_{\text{sol}}$ excludes $d_h$ **then**                         ▷ Non-exist and confident
23:         Backtrack to A2
24:     **else if** $s_{\text{sol}}$ cannot diagnose $d_h$ **then**         ▷ Exist but doubt, or Non-exist but doubt
25:         Backtrack to A3
26:     **end if**
27: **end function**

---

## B  IMPLEMENTATION DETAILS

**Evaluation Datasets.** We evaluated Med-MCTS on four medical diagnosis datasets: MedQA, Medbullets-4options, Medbullets-5options, and JMED. MedQA (Jin et al., 2020) is derived from multiple-choice questions of the United States Medical Licensing Examination (USMLE). We select disease diagnosis questions that evaluate a model's understanding and reasoning ability. MedBullets (Chen et al., 2024b) is a free learning and collaboration community that offers a large collection of USMLE style questions and study resources. The question type is primarily USMLE Step 1-style multiple-choice questions, available in both four-option and five-option versions. JMED (Wang et al., 2025b) is a novel dataset comes from JD Health's online internet hospital and is designed to simulate real clinical data. Each question includes 21 response options with a "None of the above" choice.

**Hyperparameters** We present the hyperparameters specific to Med-MCTS along with their descriptions in Table 4, while those related to the generative LLM are reported in Table 5. Regarding the weights in the multi-dimensional path evaluation, we determined them through a systematic tuning procedure: specifically, we randomly sampled 50 medical diagnosis questions from the MedQA training set to construct a small validation subset. On this subset, we performed a grid search over various combinations of the $\lambda$ parameters within the range [0.1, 1.0] ($step = 0.1$, $\sum \lambda_i = 1$). The configuration that yielded the best accuracy on this subset—$\lambda_1 = 0.3$, $\lambda_2 = 0.4$, $\lambda_3 = 0.3$—was adopted for all main experiments.

**Medical Knowledge Graph** To implement the Knowledge guided search Mechanism, we integrate a medical knowledge graph into the MCTS process to enable timely incorporation of external

Table 4: Med-MCTS hyperparameters.

| Parameter | Value |
| --- | --- |
| MCTS exploration weight | 2.0 |
| MCTS discount factor | 1.0 |
| number of rollouts | 8 |
| number of child nodes | 4 |
| maximum depth of the tree | 6 |
| maximum number of triplets retrieved from KG | 15 |

Table 5: Qwen2.5 hyperparameters.

| Parameter | Value |
| --- | --- |
| max_tokens | 1024 |
| temperature | 0.8 |
| top_k | 100 |
| top_p | 0.95 |
| num_return_sequences | 1 |

knowledge during reasoning. Specifically, we utilize an open-source knowledge graph in the medical domain, which encompasses approximately 44,000 entities and around 300,000 semantic relationships. These entities include diseases, symptoms, and examination items, with relationships describing associations such as those between diseases and symptoms, and between diseases and their etiologies.

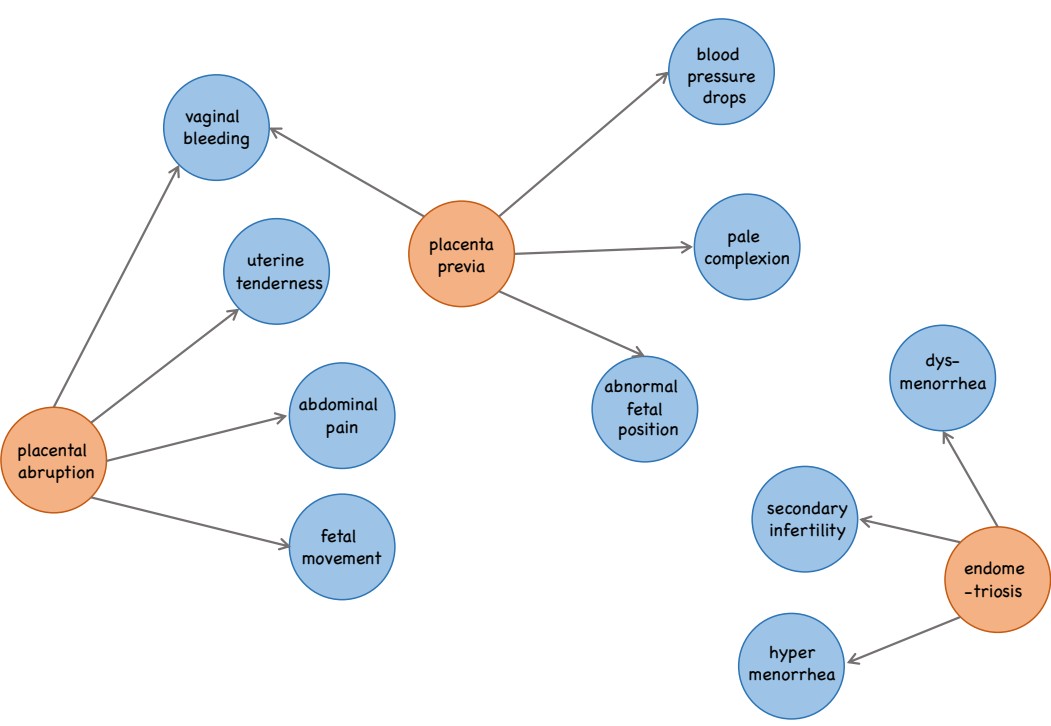

Figure 3: Medical knowledge graph of diseases and symptoms. 🟠 represents the name of disease. 🔵 refers to the symptoms or test results.

**Retrieval Corpus** For the information retrieval module, we employed the large-scale medical textbook corpus released concurrently with the MedQA dataset as our retrieval knowledge base. This design choice was motivated by several key considerations: First, medical textbooks, being

professionally edited publications, offer superior authority and accuracy. Compared to unstructured medical knowledge scraped from the internet, the textbook corpus ensures significantly higher reliability of retrieved results - a critical requirement for high-precision tasks like medical diagnosis. Second, most questions in the MedQA dataset can find direct or indirect knowledge support within the textbook corpus. This comprehensive coverage guarantees the effectiveness of our retrieval module, particularly for fundamental medical concepts and typical case analyses. In implementation, the official release provides two text segmentation approaches for the original textbook corpus: sentence-level and paragraph-level segmentation, both of which substantially enhance retrieval efficiency and accuracy.

## C  ADDITIONAL RESULTS

To further validate the source of the gains, we supplemented experiments with higher sampling budgets. We randomly selected 200 instances from the JMED dataset and generated 128 samples each using Qwen2.5-7B-Instruct and Qwen2.5-72B-Instruct. Table 6 shows the accuracy as the number of samples increases. To prevent artificial inflation of pass@k caused by output diversity in multiple-choice settings, we maintain majority voting aggregation to ensure fair comparisons. The results indicate that self-consistency slowly saturates as the number of samples grows, whereas Med-MCTS maintains a clear lead under equal or smaller sampling budgets, confirming that its improvement is not merely a by-product of deeper sampling.

Table 6: Budget-adjusted accuracy(%) performance comparison between Self-Consistency (SC) and the proposed Med-MCTS on JMED datasets.

| Model | Metric | SC (Number of Samples) | | | | | | Med-MCTS |
|---|---|---|---|---|---|---|---|---|
| | | 1 | 8 | 16 | 32 | 64 | 128 | |
| Qwen2.5-7B | Majority Vote | 52.5 | 55.5 | 54.5 | 54.0 | 54.5 | 54.5 | **58.5** |
| | Pass@$k$ | 52.5 | 70.5 | 76.5 | 80.8 | 86.0 | 90.0 | - |
| Qwen2.5-72B | Majority Vote | 59.5 | 63.0 | 63.5 | 62.5 | 62.5 | 62.5 | **67.0** |
| | Pass@$k$ | 59.5 | 81.0 | 85.5 | 89.0 | 90.0 | 93.5 | - |

## D  DISSCUSSION

**Inference Cost**  Med-MCTS enhances language models' diagnostic capabilities through test-time reasoning expansion, with its core computational overhead stemming from the reasoning tree construction process. Specifically, within the MCTS framework, the system must perform multiple rollouts (simulation samples) to explore different diagnostic pathways, a process that requires repeated calls to the underlying language model. Table 7 presents the inference costs of two model sizes—Qwen2.5-7B and Qwen2.5-72B—on the Medbullets dataset under the default configuration (8 rollouts). The table details the average number of model calls and the the number of tokens generated per question during inference. For Qwen2.5-7B, Med-MCTS requires an average of 83.5 calls, generating 168.9k tokens per question. For Qwen2.5-72B, it demands an average of 77.22 calls while producing 121.7k tokens per question. Although increasing the number of rollouts generally improves diagnostic accuracy, the computational cost grows approximately linearly. Consequently, practical applications must carefully balance performance gains against efficiency requirements.

**Computational Resources**  Our current implementation utilizes four NVIDIA A100 GPUs for deploying and running inference with the Qwen2.5-72B model. The complete experimental procedure with 8 rollouts on the Medbullets test set requires approximately 1.5 days to complete. Computational efficiency can be significantly improved through optimized batching strategies, which would substantially reduce the overall runtime.

## E  FUTURE WORK

Particularly promising is the potential to leverage search trajectory data for model self-improvement. By sampling high-quality reasoning paths to construct paired training samples, we could establish a

Table 7: Inference costs of Med-MCTS on Medbullets. We show the average number of inference calls and generated tokens required to answer one question.

|  | Qwen2.5-7B | Qwen2.5-72B |
| --- | --- | --- |
| Avg. calls | 83.5 | 77.22 |
| Avg. generated tokens | 168.9k | 121.7k |

virtuous cycle of iterative capability enhancement. This approach, combined with smarter pruning strategies and parallel computing solutions, may address current scalability limitations while maintaining the framework's diagnostic accuracy benefits. Such advancements would not only improve model's clinical utility but also expand its applicability to other domains requiring rigorous reasoning.

## F    BROADER IMPACT

While Med-MCTS enhances model's reasoning capabilities and improves both accuracy and interpretability in medical diagnostic tasks, its practical application still faces significant limitations. Due to the inherent hallucination issues and potential biases, the diagnostic results and reasoning paths generated by the model cannot yet be considered fully reliable. In the high-stakes medical domain, erroneous diagnoses could lead to serious consequences, necessitating strict regulatory oversight for any Med-MCTS-based decision support system.

## G    PROMPT TEMPLATES

---

**MedExtractor**

You are a professional medical expert skilled in extracting key medical information from unstructured patient oral descriptions. Please carefully analyze the patient's description below and extract structured information, including general features (including age, gender, medical history, etc.) and clinical features (including symptoms, test results, etc.).

The final output is in JSON format, including the following fields:
{
"General features": ["Feature 1"," Feature 2",...],
"Clinical features": ["Symptom 1"," Symptom 2",...]
"Reasoning": "Reasoning logic based on patient description (medical background analysis)"
}

### Requirement:
1. If certain information is not mentioned, the corresponding field is set to an empty list '[]'.
2. Maintain standardization of medical terminology (such as using "hypertension" instead of "high blood pressure").
3. Explain the extraction criteria in 'reasoning' (such as the patient saying 'frequently dizzy' → inferring possible 'dizziness').
4. Do not output irrelevant characters

### Patient description:
{{Patient's Verbal Description}}

---

**Action 1. Key Symptom Extraction.**

You are a skilled medical diagnostician adept at identifying the most critical and representative symptoms from patient information to prioritize potential disease hypotheses. Please carefully review the provided patient details and symptom presentation, and select the most important, urgent, and characteristic symptom(s) to focus on for formulating a plausible disease hypothesis.

The final output is in JSON format, including the following fields:
{
"Key features": ["Feature 1"," Feature 2",...],
"Reasoning": "Reasoning logic based on patient description (medical background analysis)"
}

### Requirement:
1. If certain information is not mentioned, the corresponding field is set to an empty list '[]'.
2. Maintain standardization of medical terminology (such as using "hypertension" instead of "high blood pressure").
3. Explain the criteria in 'reasoning'.
4. Do not output irrelevant characters

### Patient description:
Patient's General Features: {{Patient's General Features}}
Patient's Clinical Features: {{Patient's Clinical Features}}

**Action 2. Hypothesis Generation.**

You are a proficient medical expert skilled in developing potential disease hypotheses using patient information, symptoms, and symptom-disease relationships derived from a medical knowledge graph. Please utilize the provided details about the patient, their symptoms, and the retrieved symptom-disease associations to formulate a plausible disease hypothesis.

The final output is in JSON format, including the following fields:
{
"Hypothesis": "Disease name",
"Reasoning": "Reasoning logic based on patient description (medical background analysis)"
}

### Requirement: 1. If certain information is not mentioned, the corresponding field is set to an empty list '[]'.
2. Maintain standardization of medical terminology (such as using "hypertension" instead of "high blood pressure").
3. Explain the criteria in 'reasoning'.
4. Do not output irrelevant characters

### Patient description:
Patient's General Features: {{Patient's General Features}}
Patient's Key Features: {{Patient's Key Features}}

### Triples retrieved from the knowledge graph
{{Retrieved Symptom-Disease Triples from Knowledge Graph}}

## Action 3. Evidence Verification.

You are a proficient medical expert skilled in identifying additional relevant symptoms to investigate further based on patient information, symptoms, existing disease hypotheses, and symptom-disease relationships derived from a medical knowledge graph. Please utilize the provided details about the patient, their symptoms, the current disease hypothesis, and the retrieved disease-symptom associations to formulate a plausible next step for further symptom inquiry.

The final output is in JSON format, including the following fields:
{
"Relevant symptom": "symptom name",
"Reasoning": "Reasoning logic based on patient description (medical background analysis)"
}

### Requirement:
1. If certain information is not mentioned, the corresponding field is set to an empty list '[]'.
2. Maintain standardization of medical terminology (such as using "hypertension" instead of "high blood pressure").
3. Explain the criteria in 'reasoning'.
4. Do not output irrelevant characters

### Patient description:
Patient's General Features: {{Patient's General Features}}
Patient's Key Features: {{Patient's Key Features}}
Current disease hypothesis: {{Current Hypothesis}}

### Triples retrieved from the knowledge graph
{{Retrieved Disease-Symptom Triples from Knowledge Graph}}

> **Action 4. Deductive Analysis.**
>
> You are a proficient medical expert skilled in evaluating patient symptoms and clinical indicators to validate or refute potential disease hypotheses. Please utilize the provided patient's oral description, the disease hypothesis, and the relevant clinical indicator to be verified to determine the presence of the symptom and its impact on confirming or refuting the disease hypothesis.
>
> The final output is in JSON format, including the following fields:
> {
> "Existence": "Exist/Non-exist",
> "Certainty": "Confident/Doubt",
> "Reasoning": "Reasoning logic based on patient description (medical background analysis)"
> }
>
> The final result should fall into one of the following categories:
> Exist and Confident: The clinical indicator exists and supports the current hypothesis, leading to a definitive diagnosis.
> Exist but Doubt: The clinical indicator exists, but uncertainty remains. Further diagnostic tests are needed to confirm the hypothesis.
> Non-exist and Confident: The clinical indicator does not exist, refuting the current hypothesis. Other potential diseases should be considered.
> Non-exist but Doubt: Although the clinical indicator does not exist, other clinical findings or ancillary information may still support the current hypothesis. Further pathological analysis is needed to explain the discrepancy.
>
> ### Requirement:
> 1. If certain information is not mentioned, the corresponding field is set to an empty list '[]'.
> 2. Maintain standardization of medical terminology (such as using "hypertension" instead of "high blood pressure").
> 3. Explain the criteria in 'reasoning'.
> 4. Do not output irrelevant characters
>
> ### Patient description:
> Patient's Verbal Description: {{Patient's Verbal Description}}
> Current Disease Hypothesis: {{Current Hypothesis}}
> Clinical Indicator to be Verified: {{Clinical Indicator to be Verified}}

# H GENERATIVE AI STATEMENT

We acknowledge the use of generative AI in this work. Specifically, we employed LLMs to provide editorial support during the preparation of the manuscript.

