# OpenReview forum: "Knowledge-enhanced MCTS for LLM-based Medical Diagnosis Reasoning"
_ICLR.cc/2026/Conference — ICLR 2026 Conference Withdrawn Submission_

### Official Review · Reviewer_rJoy · 2025-10-29

**Soundness:** 3
**Presentation:** 3
**Contribution:** 3
**Rating:** 6
**Confidence:** 2

**Summary:**

This paper introduces Med-MCTS, a diagnostic reasoning framework that integrates Monte Carlo Tree Search (MCTS) with a medical knowledge graph. The core of Med-MCTS features a clinically inspired action space (A1–A4: key symptom extraction, hypothesis generation, evidence verification, and deductive analysis) that closely aligns with physicians’ iterative diagnostic workflows. A bidirectional knowledge retrieval mechanism (R1: symptom-to-disease, R2: disease-to-symptom) provides structured and verifiable knowledge support during tree expansion and verification. Additionally, a multi-dimensional path discriminator—encompassing Consistency, Diversity, and Agent Evaluation—selects high-quality reasoning paths, replacing simple majority voting. Across benchmarks including MedQA, MedBullets, and JMED, Med-MCTS outperforms baselines such as CoT, RAG, Self-Consistency, RAP, and rStar on both Qwen2.5-7B and 72B models. Compared to domain-specific models, it achieves higher overall accuracy and approaches GPT-4o performance. Ablation studies highlight the substantial contributions of R1/R2, A4, and discriminator.

**Strengths:**

-  The paper adapts the Monte Carlo Tree Search (MCTS) to the domain of clinical diagnosis by introducing a clinically-inspired action space (A1–A4). It explicitly models backtracking forks based on the presence/absence and certainty/uncertainty of symptoms, which distinguishes it from conventional mathematical decomposition approaches.
-  The paper is well-written and clearly organized, following a logical structure (Method, Experiments, Ablation, Limitations). The framework's workflow is effectively illustrated with diagrams (Figures 1 and 2), and technical details are made accessible through clear mathematical formulas (for UCT and scoring) and pseudocode (Algorithm 1).

**Weaknesses:**

- Lack of a Clear Reward Definition for MCTS: The paper states that it uses the UCT algorithm, which requires backpropagating a value Q(s,a), but it fails to explicitly define how the terminal reward for a rollout is calculated. It is unclear whether this reward is a binary score based on the ground-truth answer, a value derived from the model's confidence, or a scalar score from the multi-agent evaluator. This omission is a critical gap in the methodology that hinders reproducibility.
- Insufficient Detail in the Multi-dimensional Scorer: The path selection mechanism is not fully specified. Specifically: (a) The implementation of the similarity function Sim(·,·) for calculating diversity is not described. (b) The acronym 'MAE' for 'Multi-Agent Evaluation' is ambiguous and can be easily confused with 'Mean Absolute Error'. (c) The paper only briefly mentions how the sub-scores are normalized and how their weights are tuned, lacking clear details on the measurement scales or calibration methods.
- Ambiguity Regarding the Retrieval Corpus: The paper's methodology focuses on knowledge graph (KG) retrieval, yet the appendix describes a "Retrieval Corpus" based on textbook materials. It is not clearly distinguished whether this text-based corpus is used exclusively for the RAG baseline or if it is also leveraged by the proposed Med-MCTS framework. This creates confusion about the precise knowledge sources used by the method.
- Presentation and Referencing Errors: The paper contains several distracting presentation issues. There are multiple spelling errors (e.g., "Accuray," "Mult-Agent evalution," "selecte") and an incorrect cross-reference in Section 3.2, where the text describing the general MCTS structure points to Figure 2 (a specific case study) instead of a more general diagram.

**Questions:**

Please clarify the definition and value range of the reward used for backpropagation. Is it a binary correctness score (e.g., 1 for a correct terminal diagnosis, 0 otherwise), a value mapped from the confidence level in action A4, or a scalar score from the multi-agent evaluation?

What is the specific implementation of the Sim function (e.g., text edit distance, cosine similarity of sentence embeddings, structural alignment)? For the 'MAE' score, what is its calibration and range (e.g., [0, 1]), and are the individual sub-scores normalized?

Could you provide more complete results for the hyperparameter tuning of λ, such as the grid search outcomes or a sensitivity analysis with confidence intervals?

Regarding the "Retrieval Corpus" described in the appendix: is the textbook retrieval used exclusively for the RAG baseline, or does Med-MCTS also utilize text retrieval during certain action phases?

---

### Official Review · Reviewer_TU3q · 2025-11-01

**Soundness:** 3
**Presentation:** 3
**Contribution:** 2
**Rating:** 4
**Confidence:** 4

**Summary:**

The paper proposes Med‑MCTS, a test‑time reasoning framework that combines Monte‑Carlo Tree Search (MCTS) with knowledge‑grounded retrieval for clinical diagnosis with LLMs. The key ideas are:

* A clinically informed action space that mirrors physician workflow: (A1) Key Symptom Extraction -> (A2) Hypothesis Generation -> (A3) Evidence Verification -> (A4) Deductive Analysis with explicit backtracking rules.
* Knowledge‑grounded search using a medical knowledge graph (about  44k entities / about 300k relations) with bidirectional retrieval: R1 (symptom -> disease) for hypothesis generation and R2 (disease -> symptom) for targeted verification, plus a similarity threshold \tau to gate KG use.
* A multi‑dimensional path evaluator selecting the final diagnosis by combining consistency*, *diversity (pairwise path similarity), and agent‑based factuality.

Results. On four MCQ benchmarks (MedQA, MedBullets‑4/5 options, JMED) and two backbones (Qwen2.5‑7B/72B‑Instruct), Med‑MCTS improves over CoT, RAG, Self‑Consistency, RAP, and rStar. For Qwen2.5‑72B, it reaches 71.10/62.66/84.57/68.00% on the four datasets, outperforming domain‑specific medical models and approaching GPT‑4o.

Ablations & cost. Removing KG retrieval (R1/R2) or A4 reduces accuracy; the full system performs best on a MedQA subset (72.5%). Inference is compute‑heavy (~77–84 LLM calls; 121k–169k tokens/question).

**Strengths:**

1. Well‑aligned action space that mirrors real diagnostic workflow (A1–A4), enabling interpretable search and principled backtracking.
2.  Knowledge‑grounded expansion via KG (R1/R2), addressing the limitations of surface‑similarity RAG and strengthening factual grounding.
3. Consistent empirical gains across datasets and model scales; Med‑MCTS with 72B outperforms domain‑specific medical models and is competitive with strong general LLMs.
4. Ablations and budget study support claims that gains are not merely from heavier sampling; increased pass@k for SC saturates while Med‑MCTS maintains an edge under similar budgets.
5. Transparency and practicality: the tree exposes "show‑your‑work" reasoning; a small expert study (50 cases) reports good correctness and interpretability (about 4.1 – 4.2 / 5).

**Weaknesses:**

1. Evaluator transparency: The diversity metric and multi‑agent evaluation (MAE) are not fully specified (models/prompts; whether evaluators see the final answer; safeguards against self‑confirmation). This is critical because the final selection depends on Eq. (4).
2. Reproducibility:

    * The KG identity, construction, licensing, and access are not named; τ (similarity threshold) and embedding model (f_\theta) are unspecified. Without these, re‑running R1/R2 is difficult.
    * Hyperparameters for the scorer (\lambda) were tuned on a 50‑question MedQA subset; robustness across datasets is not analyzed.
3. Statistical rigor: No CIs or significance tests for deltas over rStar/RAP; some gains are modest (~=1-2%).
4. Compute overhead: The approach is expensive at inference (~77–84 calls; up to ~169k tokens/question), which limits deployability; no adaptive‑budget or early‑exit mechanism is presented.
5. Evaluation scope: All tasks are multiple choice; while JMED is larger‑option (21) and more realistic, fully open‑ended differential diagnosis or free‑text synthesis/citation tasks are not reported.
6. Fairness of baselines: The RAG baseline uses a textbook corpus; Med‑MCTS uses a KG. The paper argues KG superiority, but a matched data source (e.g., KG derived from the same textbooks) or RAG + KG hybrid baselines would strengthen the claim.

**Questions:**

1. Evaluator specifics:

   * How exactly is Sim(t_i,_j) computed (embedding space? sentence‑level cosine? structure‑aware)?
   * For MAE, which models, prompts, and verifiers are used? Do verifiers see the final answer choices or only paths? Any adjudication tie‑breaks?
2. KG details and reproducibility:
   * Which knowledge graph (name/source) is used? Will you release the KG (or mapping layer) to enable replication? What is the value of τ and the exact embedding model (f_\theta)?
3. Statistical reporting: Could you provide 95% CIs or bootstrap tests for Table 1/2 deltas (especially vs rStar/RAP), and per‑dataset error bars?
4. Ablations beyond MedQA: The ablation in Table 3 uses 120 MedQA samples. Can you replicate on JMED or MedBullets to show robustness of R1/R2 and A4 across distributions?
5. Efficiency / scaling: Any results with adaptive rollouts (e.g., early exit when posterior concentrates), dynamic branching, or UCT parameter sweeps to trade accuracy vs cost?
6. Failure analysis & safety: Please include a qualitative error analysis (common failure modes) and a calibration or hallucination check for reasoning paths, beyond the small expert rating.
7. Fair baseline comparison: Could you run a KG‑only RAG (no MCTS) and/or RAG+KG with the same KG to isolate the benefit of MCTS vs data source?
8. Even though you leverage MCTS and KG to imporove factuality during generation, how could you audit the factuality of the internal process? It resembles infinite regress (aka, "Turtles all the way down"). Please elaborate this.

---

### Official Review · Reviewer_pBcE · 2025-11-01

**Soundness:** 2
**Presentation:** 1
**Contribution:** 2
**Rating:** 2
**Confidence:** 3

**Summary:**

The paper presents a method for medical diagnosis with external knowledge, which builds a tree whose nodes are states and edges are actions (the nodes are not explained in the paper). A diagnosis is done by expanding the tree until reaching a specific terminal diagnosis. The experimental results show that the performance of the method is better than some baseline models, but it did not outperform recent LLMs.

**Strengths:**

1. The motivation of the work makes sense.

**Weaknesses:**

1. It’s hard to see what is done in the method. This may be because some mathematical notation is not defined. For example, $\mathcal{S}$ and $Q$ in Eq. (1) are not defined.
2. Also, I don’t see how the external knowledge is used in the method. It should somehow interact with the tree in Figure 1, but I cannot see it in the main text and Algorithm 1 does not fully explain it.
4. I can find a paper which seems relevant to the paper [Wang et al., “DiReCT: Diagnostic reasoning for clinical notes via large language models,” NeurIPS 2024]. The paper seems to use a graph to specify the deductive process until the terminal diagnosis explicitly. It would be nice to see a task-level, methodological, and experimental comparison with the paper.
5. It would be desirable to show how the identified path for a diagnosis makes sense for human experts or some qualitative evaluation on this point.

**Questions:**

1. For me, as the medical domain often uses special vocabulary, surface-level retrieval seems to work well. Is there an ablation study on this point?
2. What is $S$ in Eq. (2)?

---

### Official Review · Reviewer_fWYC · 2025-11-02

**Soundness:** 2
**Presentation:** 2
**Contribution:** 2
**Rating:** 2
**Confidence:** 5

**Summary:**

This paper proposed Med-MCTS, a knowledge-enhanced diagnostic reasoning framework that integrates MCTS with external medical knowledge.

**Strengths:**

This whole framework is reasonable, which is aligned with the real disease diagnosis. It seems to be easily combined with any LLMs to improve their reasoning ability from Table1.

**Weaknesses:**

1. I’m curious about the accuracy of retrieving entities using embedding similarity. You mentioned the limitations of RAG in the paper, but I wonder whether cosine similarity can reliably retrieve the correct entities. Why not use a more rigorous retrieval strategy?
2. What happens if there are no similar entities found in the knowledge graph? I am also very confused on the hypothesis generation. No matter the symptoms or prescribed medications or patient populations, they are recorded in clinical practice guidelines or drug label. Why do we need a hypothesis generation module? Even if only with KG, you could reason on graph directly. Is this necessary?
3. Does the model’s performance depend on which knowledge graph is used?
4. I would like to see a comparison between Med-MCTS and a KG-based RAG method.
5. What is the performance of pure LLMs? Please show the results of Qwen2.5-72B-Instruct when prompted directly. I’m also curious why Med-MCTS does not outperform other models of similar size, such as HuatuoGPT-o1-72B.
6. Overall, the framework seems to lack novelty. It looks like a method that samples reasoning traces and then selects one based on the knowledge graph. The performance is highly depended on adopted knowledge graph.

**Questions:**

see weaknesses

---

### Note · Authors · 2026-01-09

I have read and agree with the venue's withdrawal policy on behalf of myself and my co-authors.